# Effectiveness of a Nurse-Led Telehealth Intervention to Improve Adherence to Healthy Eating and Physical Activity Habits in Overweight or Obese Young Adults

**DOI:** 10.3390/nu16142217

**Published:** 2024-07-11

**Authors:** R. García-Rodríguez, A. Vázquez-Rodríguez, S. Bellahmar-Lkadiri, A. Salmonte-Rodríguez, A. R. Siverio-Díaz, P. De Paz-Pérez, A. M. González-Pérez, P. R. Brito-Brito

**Affiliations:** 1La Laguna-Tenerife Norte Multiprofessional Family and Community Care Teaching Unit, Primary Care Management Board of Tenerife, The Canary Islands Health Service, 38003 Santa Cruz de Tenerife, Canary Islands, Spain; rociogr.efyclalaguna@gmail.com (R.G.-R.); arantxavr.efyclalaguna@gmail.com (A.V.-R.); samiabl.efyclalaguna@gmail.com (S.B.-L.); albertosr.efyclalaguna@gmail.com (A.S.-R.); albasd.efyclalaguna@gmail.com (A.R.S.-D.); patriciapp.efyclalaguna@gmail.com (P.D.P.-P.); agonzalp@ull.edu.es (A.M.G.-P.); 2Nursing Department, Faculty of Healthcare Sciences, University of La Laguna, 38200 Santa Cruz de Tenerife, Canary Islands, Spain

**Keywords:** diet, healthy, exercise, nursing care, overweight, obesity, telemedicine, primary health care, standardised nursing terminology

## Abstract

Excess weight and obesity have high prevalence rates globally and are associated with significant morbidity and mortality rates. In the wake of the COVID-19 pandemic, telehealth emerged as an effective tool for promoting healthy behaviours in the management of chronic conditions. This study employed a three-month quasi-experimental design with pre- and post-intervention measurements, assessing the progress of 82 adults assigned either to an intervention group (IG) or a control group (CG). A total of 66 participants completed the study, with 33 in each group. The IG had access to a telehealth-based platform providing educational content on healthy lifestyle habits and were followed up both in-person and remotely. The CG were monitored as usual, i.e., at their primary care nurses’ discretion. The IG exhibited a statistically significant reduction in weight, BMI, and abdominal circumference, along with improved adherence to a heart-healthy diet, specifically the Mediterranean diet, and higher levels of physical activity. The nurse-led intervention (Care4US+), utilising telehealth from primary care, has proven effective in enhancing healthy behaviours and reducing cardiovascular risk factors.

## 1. Introduction

The high prevalence of excess weight and obesity, along with their associated morbidity and mortality rates in both developed and developing countries, has led to the recognition of these issues as major global public health concerns [1]. Since 2020, the prevalence of excess weight and obesity has been exacerbated by the SARS-CoV-2 pandemic due to decreased physical activity and increased sedentary behaviour [2,3]. 

In Spain, according to the 2020 European Health Interview Survey, 45% of men and 30% of women over the age of 18 are overweight, while 17% of men and 16% of women are obese [4]. Furthermore, it is estimated that weight-related disorders result in a global economic burden of approximately USD two trillion, equating to 2.8% of the world’s GDP [5]. In Spain, these health issues incur an additional direct cost of EUR two billion to the National Health System [6].

In addressing weight issues and their complications, nurses possess a theoretical framework applicable to the assessment, diagnosis, and treatment of excess weight and obesity, i.e., a middle-range theory described by authors Pickett, Peters, and Jarosz in ‘Toward a Middle-Range Theory of Weight Management’. This theory adapts Orem’s self-care deficit model to weight control and management [7]. Additionally, the NANDA-I association currently includes these issues among its nursing diagnoses: Risk for Overweight (code 00234), Overweight (code 00233), and Obesity (code 00232) [8]. Within the context of standardised nursing languages, these diagnoses are linked to numerous health outcomes criteria and nursing interventions, as classified in the Nursing Outcomes Classification (NOC) and the Nursing Interventions Classification (NIC) taxonomies, respectively [9,10].

In the NANDA-I classification, there are other nursing diagnoses useful for addressing these issues, known as health promotion nursing diagnoses, which focus on motivating individuals to make favourable changes in their dietary and exercise behaviours. These nursing diagnoses include Readiness for Enhanced Nutrition (code 00163) and Readiness for Enhanced Exercise Management (code 00307) [8].

Currently, digitalisation is one of the burgeoning strategies for managing and improving health. The use of information and communication technologies has become widespread, to the extent that concepts like telehealth, tele-nursing, or digital health are commonplace. Telehealth has been rapidly implemented since the COVID-19 pandemic to ensure patient care, although there are limited data on how this shift in care models may have impacted outcomes and interventions for weight loss in overweight or obese patients.

In 2022, a comparative and retrospective observational study was published with the aim of analysing weight changes following different consultation approaches [11]. Some of these involved in-person visits, while others combined in-person visits with teleconsultations via video calls. A third method consisted solely of video calls. Weight changes over the six-month follow-up period did not significantly differ between the groups. Therefore, the inclusion of new technologies, compared to traditional in-person care, has not resulted in a loss of effectiveness in weight reduction efforts. 

It is worth noting that telehealth involves more than just changing the communication channel or space, as information and communication technologies also enable health education and training, promote greater user participation, and facilitate monitoring of patients’ progress [12].

Another study conducted in 2021 demonstrated the effectiveness of a telehealth-based intervention combining automated text messages, use of wearable telemonitoring devices, and health counselling to promote healthy lifestyle changes [13]. Over 16 weeks, the study targeted 30 overweight or obese adult patients with moderate-to-vigorous physical activity levels of less than 150 min per week, aiming for weight loss. The study resulted in a significant mean weight loss of nearly 4 kg and a decrease in body mass index (BMI) of 1.8 kg/m^2^.

Moreover, a randomised clinical trial demonstrated the effectiveness of a motivational intervention utilising information and communication technologies [14]. This study followed 185 overweight or obese patients in a primary care facility over one year. Three groups were established, each receiving a distinct intervention approach. Group 1 received standard weight loss recommendations; group 2 underwent a motivational intervention coordinated by a trained nurse; and group 3 received the same motivational intervention combined with a digital platform for recording, monitoring, and promoting physical activity. Participants across all groups showed weight reduction, with statistics showing significantly greater weight loss observed in group 3. This reduction in weight was accompanied by decreased BMI and cholesterol levels. The study concluded that integrating a digital platform to encourage physical activity alongside motivational intervention in overweight or obese patients provides substantial additional benefits in terms of weight loss, BMI reduction, and lipid profile improvement at a reduced cost.

Another randomised clinical trial was conducted to study whether a telehealth-based lifestyle counselling programme led to greater weight loss over a period of 12 months compared to standard care [15]. The study included 340 individuals with obesity, where the intervention group received an initial face-to-face motivational interview followed by personalised online counselling using behavioural techniques. Participants in the intervention group received counselling based on individualised care plans, being congratulated after achieving each goal to maintain high levels of motivation. The primary outcome was a significant decrease in mean body weight and BMI in both groups, with a greater reduction observed in the intervention group. It was thus concluded that planned digital lifestyle counselling can lead to greater weight loss compared to standard care. Given the results of studies as described, telehealth represents an opportunity to promote patient empowerment and self-care. 

The general objective of this study was to improve the management of excess weight and obesity in a sample of adults visiting nurses’ offices in primary care facilities following a telehealth-based educational intervention. The specific objectives were: (1) to develop a telemedicine intervention strategy for the treatment of grade I–II excess weight and obesity by leveraging the opportunities offered by information and communication technologies; (2) to achieve a significant improvement in the reduction of somatometric indicators; (3) to increase participants’ levels of physical activity and exercise; and (4) to enhance their skills and attitudes towards a healthy lifestyle, as well as their satisfaction with telehealth.

## 2. Materials and Methods

### 2.1. Design and Sampling Method

We conducted a quasi-experimental study to measure the effectiveness of a telehealth-based nursing intervention employing pre–post measurements with a prospective, analytical design and quota-based allocation to each group, the intervention group (IG) and control group (CG). The study targeted overweight or obese adults attending nurses’ offices in primary care facilities within the Tenerife Healthcare Area (Canary Islands, Spain). 

The inclusion criteria included: (a) individuals aged 18–45 years; (b) who are overweight or obese (grade I–II) based on BMI; (c) willing to participate in the study through informed consent; (d) motivated to change according to the Prochaska and DiClemente model; and (e) attending nurses’ offices in primary care facilities in Tenerife during the study period. To assess each potential participant’s readiness for change, they have to be in the preparation or action stage according to the Transtheoretical Model of Change by Prochaska and DiClemente, during initial assessment by the nurse [16]. Exclusion criteria encompassed individuals with chronic health conditions that prevent participation in the intervention, especially involving moderate-intensity physical activity or significant mobility limitations. 

The initial estimate suggests that an approximate sample size of 100 participants will suffice to achieve the planned objectives of this research for comparative measurement between both groups. A sample size close to the aforementioned figure, 100 cases, would allow us to estimate nonparametric correlation coefficients of at least 0.28 and to verify the expected differences in the before-after comparisons, with a minimum output of 90% in two-tailed hypothesis testing and at an alpha significance level of 0.05, a 5% margin of error and 95% confidence levels. Participants were randomly assigned based on their recruitment moment, alternating between the IG and the CG until the total required sample size was reached. To qualify, participants had to meet all inclusion criteria and none of the exclusion criteria, as well as agree to participate after being informed about the study characteristics and signing the informed consent form.

### 2.2. Study Setting

The Tenerife Healthcare Area is one of seven regions comprising the Canary Islands Health Service, which is integrated into the Spanish National Health System and provides public healthcare across the Canary Islands. As of 2023, this Spanish Autonomous Community had a total population of approximately 2,178,000 inhabitants, with around 932,000 residing specifically on the island of Tenerife [17]. The Tenerife Healthcare Area is subdivided into 41 Basic Healthcare Districts, encompassing a total of 103 primary care facilities, where 800 nurses deliver primary healthcare services. This study employed random selection from three Basic Healthcare Districts in the northern geographical area of the island, serving a combined total of 86,997 residents through 35 nurses’ offices. Within this population, 18.7% (16,305 individuals) had been diagnosed with excess weight conditions as documented in their electronic health records. 

### 2.3. Construction and Design of the Data Collection Notebooks

Two types of data collection notebooks were created. The first one, for initial or baseline assessment purposes, included sociodemographic variables (sex and age), anthropometric variables (weight, abdominal circumference, and BMI), motivational variables, and variables related to dietary and physical activity habits. To assess initial motivation, a Likert scale (1 to 5, from lowest to highest) was included where individuals subjectively indicated their level of motivation to lose weight at the baseline moment. Dietary habits were assessed using the Cuestionario de Adherencia a la Dieta Mediterránea [the Mediterranean Diet Adherence Questionnaire] or CADM, consisting of 14 items with dichotomous responses and possible scores ranging from 0 to 14, representing poorer to better adherence [18]. Physical activity was assessed using the International Physical Activity Questionnaire (IPAQ), which comprises seven items that evaluate the quantity and intensity of physical activity performed during the previous week [19]. For the purposes of this study, the metabolic expenditure for walking and moderate and/or vigorous physical activity was calculated in MET units (1 kcal/kg/h). The second data collection notebook, for post-intervention assessment, consisted of the same sets of variables as the initial notebook, excluding motivational variables and adding a final satisfaction assessment containing three questions related to the intervention, rated from 1 to 5, from lowest to highest satisfaction. These questions were: 1. To what extent are you satisfied with the use and availability of the telehealth audiovisual resources employed when receiving the intervention? 2. How satisfied are you with the quality of the content of the intervention you received? 3. How satisfied are you overall with the educational intervention for the promotion of better dietary and physical activity habits?

### 2.4. Design and Preparation of the Nursing Intervention 

A three-month individual nursing intervention on a hybrid basis (both remote and in-person appointments), termed Care4US+, was designed based on the objectives of the middle-range theory for weight management and the following standardised interventions from the NIC classification [10]: Teaching: Prescribed Exercise (code 5612), Exercise Promotion (code 0200), Teaching: Prescribed Diet (code 5614), and Behaviour Modification (code 4360). The nursing diagnoses that identify good motivation or readiness for improvement in dietary and physical exercise habits—summarised as enhancing adequate self-care of their health—are documented in the patients’ care plans in their electronic health records. These NANDA-I nursing diagnoses are: Readiness for Enhanced Health Self-Management (code 00293), Readiness for Enhanced Nutrition (code 00163), and Readiness for Enhanced Exercise Engagement (code 00307). 

The nursing assessment of all study participants (IG and CG) was conducted during an initial face-to-face consultation (at the time of each patient’s inclusion in the study), as well as during a final face-to-face consultation to collect the necessary information. The CG received standard care with exclusively face-to-face follow-ups prescheduled by their respective family care units. The IG underwent a series of intermediate follow-up consultations, both face-to-face and via telephone or email, according to individual preferences and needs, with continuous access to the telehealth materials created for the web space, which were accessible from any mobile device or computer with internet access. 

The intervention content was structured into three blocks: 1. Healthy Eating; 2. Recommended Physical Exercise; and 3. Motivation. The work materials for each block were made available to IG participants on a web space that is part of an e-learning platform belonging to the institution conducting the research. Each content block includes various audiovisual materials, such as infographics, short videos, links to relevant websites, and brief questionnaires on basic concepts. The objective of these materials, and the intervention as a whole, is to promote the improvement of dietary and physical activity habits to reduce weight through motivation and positive reinforcement. 

### 2.5. Data Collection Procedure

The study was approved in May 2022 by the Provincial Ethical Committee for Research with medicinal products under reference code CHUC_2022_61. An active recruitment phase was then planned, lasting one month, targeting patients from the Basic Healthcare Districts included in the study and inviting all potential candidates to participate. Telephone calls were made to 388 potential participants. After recruiting 101 individuals who agreed to participate in the study, appointments for the initial assessment in the nurses’ offices were scheduled. Ultimately, 82 individuals attended the assessment, completed the baseline data collection notebook, and were assigned to either the IG or CG after providing informed consent. The field phase, which involved both the experimental intervention (IG) and conventional care by their family care units (CG), was conducted from 15 December 2022 to 15 March 2023. Upon concluding the intervention phase, the data were entered into an Excel^®^ v.16.78.3. database which was later exported to the SPSS^®^ v.29.0. statistical program for cleaning and analysis. 

### 2.6. Data Analysis

The sample was described using frequency percentages for categorical variables, and either means and standard deviations for scale variables if the sample distribution was normal, or median and interquartile ranges if it was non-normal. The normality of the distribution for scale variables was analysed using the Kolmogorov–Smirnov test. To compare pre–post changes in the variables targeted by the intervention, Student’s *t*-test or the Wilcoxon test—depending on the normality of the distribution—and McNemar’s test were used. All tests were two-tailed, with a statistical significance threshold of *p* < 0.05, conducted using the SPSS^®^ v.25.0 statistical program.

### 2.7. Ethical Considerations

The data collection notebooks included fully anonymised information, with no possibility of identifying participants, as each was assigned a numerical code. The study was conducted in compliance with current European and Spanish laws and regulations regarding the processing, communication, and transfer of personal data, adhering to the provisions of Regulation (EU) 2016/679 of the European Parliament and of the Council of 27 April 2016 (GDPR) and the Spanish Organic Law 3/2018 of 5 December, on Personal Data Protection and the Guarantee of Digital Rights. To ensure the confidentiality of the study participants, only the research team, or the Ethics Committee and Health Authorities if required, had access to the data. 

## 3. Results

Out of the 82 participants who attended the initial assessment, 16 dropped out of the study, with 5 from the CG and 11 from the IG. No significant differences in sociodemographic and clinical characteristics were observed between those who dropped out and those who did not. A total of 66 participants completed the study, with 33 in each group. 

Regarding the evaluation of differences between patients assigned to the CG or the IG, a significantly lower mean age was observed in the IG (*p* = 0.013). 

### 3.1. Description of Both Groups (CG and IG) after the Initial Assessment 

Table 1 shows the main characteristics summarising the information of both groups (CG and IG) at the time of each patient’s initial assessment.

### 3.2. Pre–Post Comparison of Anthropometric Characteristics, Adherence to the Mediterranean Diet, and Physical Activity Level between the CG and the IG

Table 2 shows the differences in weight, abdominal circumference, and BMI, as well as the scores obtained in the CADM and metabolic expenditure between the CG and IG at the baseline and final points.

A significant decrease in weight, abdominal circumference, and BMI was observed in the IG, along with a significant increase in BMI in the CG. Additionally, the IG achieved a significant improvement in adherence to the Mediterranean diet and a significant increase in total metabolic expenditure.

### 3.3. Pre–Post Comparison of Percentages and Number of Correct Responses between the CG and IG for Each Item of the Mediterranean Diet Adherence Questionnaire (CADM)

Table 3 describes the frequency of correct responses regarding dietary habits for each CADM item at the pre- and post-intervention points according to the participants’ group (CG or IG).

As observed in Table 3, both groups showed improvements in adherence to the Mediterranean diet. In the CG, nine of the items showed increased percentages of correct responses, with smaller differences compared to the IG. Additionally, in the CG, the percentages for 4 items remained the same, and only one item decreased (item 9). In the IG, higher frequencies of correct responses were achieved post-intervention for all CADM items with the exception of item 8, with significant differences for items 3, 4, 5, and 7. 

### 3.4. Pre–Post Comparison by Age Groups of Anthropometric Characteristics, Adherence to the Mediterranean Diet, and Physical Activity Level between the CG and IG 

Table 4 shows the differences between the CG and IG by age groups (18–26; 27–35; and 36–45 years) for the variables weight, abdominal circumference, BMI, CADM scores, and metabolic expenditure as measured by the IPAQ at the baseline and final points. 

This age-group analysis revealed significant improvements in adherence to the Mediterranean diet for all three categories in the IG. For the 27–35 years age group, significant reductions in weight and BMI were also observed in the IG. Additionally, for the 36–45 years category, higher metabolic expenditure was noted in the IG. Conversely, in the CG, a significant increase in weight and BMI was found in the 36–45 years age group.

### 3.5. Satisfaction of the IG with the Received Intervention

Table 5 presents the satisfaction of IG participants with the received intervention for improving their dietary and physical activity habits at the final point of the study. 

For satisfaction question 1, related to the use and availability of telematic resources, more than half of the IG participants (54.5%) were very satisfied or completely satisfied. Regarding questions 2 and 3, associated with the quality of content and overall satisfaction, 96.9% in both cases reported being very or completely satisfied.

## 4. Discussion

This study successfully achieved the primary objective of improving the management of excess weight and obesity (grades I–II) in the IG following a telehealth-based educational intervention. The intervention lasted three months and was conducted individually with patients attending nurses’ offices in primary care facilities. The semi-presential format provided participants with content to enhance healthy lifestyle habits and addressed their individual needs through further follow-up appointments, phone calls, and emails. Our results align with those of other studies [13,14,15], confirming the benefits of interventions based on telehealth and participant motivation. New studies highlight the benefits of including family members in interventions to promote weight loss, maintaining common goals and fostering family unity [20], which could be relevant for primary care and future research in this field. 

It is well known that the global prevalence of obesity is higher among women [21] and that, in our study context, they perceive their health as good or very good less frequently than men [22]. Among the participants in both the CG and the IG, there was a majority of women, about 7 out of 10, similar to other studies mentioned above [14,15]. This may be partly due to a greater initiative among women to participate in weight reduction programmes, as women are known to be more engaged in this field and display better levels of health self-care [23]. Women also place high importance on weight control and body image due to prevailing psychological, social, and cultural factors [24], which is further emphasised by the influence of social media [25]. 

The inclusion criteria for this study focused on individuals aged 18–45 years, as habit changes are recommended for younger age groups to achieve sustainable results and better quality of life over time [26]. The results for the IG as a whole (Table 2) show significant reductions in weight, waist circumference, and BMI, as well as increased adherence to the Mediterranean diet and physical activity. In the CG, the only significant change observed was an increase in BMI. Our intervention targeted a group of younger individuals (Table 1), which could introduce a selection bias due to chance. However, segmented analysis by age range within the IG (Table 4) showed no better outcomes for the 18–26 age group compared to the other groups (27–35 and 36–45). All three groups demonstrated improvements in dietary habits, but the 27–35 age group also achieved weight and BMI reductions, and the 36–45 age group increased their physical activity, all of which were statistically significant. 

When comparing pre- and post-intervention changes between the CG and IG using the CADM, significant differences were found in items 3, 4, 5, and 7 (Table 3). These items reflect better adherence in the IG regarding the consumption of vegetables, fruits, red and processed meats, and carbonated or sugary drinks. For the remaining items, the IG also showed better adherence except for wine consumption. The CADM positively evaluates the intake of three or more glasses of wine per week. Thus, participants might misinterpret the item, as consuming three glasses or fewer is rated unfavourably. Additionally, there is controversy regarding safe alcohol intake, and no consumption should be recommended due to the lack of evidence supporting a safe threshold [27]. 

The findings of this quasi-experimental study validate the effectiveness of a semi-presential nurse-led intervention like Care4US+, based on information and communication technologies and grounded in its own conceptual framework [7] for weight management in individuals ready for change. This approach utilises standardised nursing terminology from NANDA-I, NOC, and NIC, which contribute to better describing care realities and achieving positive practice outcomes [28]. The intervention method aligns with other health education strategies in similar populations [29] or even in other profiles of patients who, in addition to obesity, suffer from hypertension [30] or multiple sclerosis [31] or have undergone bariatric surgery [32]. These strategies, including Care4US+, use continuous motivational techniques and engaging audiovisual resources, easy to understand and accessible through a web-based e-learning platform. This likely leads to high participant satisfaction, as observed in our IG results (Table 5). 

This study has a number of limitations. Firstly, the sample size completing participation was 66 across both groups, CG and IG. This sample size can limit the generalizability of the findings. Other telehealth-based studies for weight reduction and physical activity promotion in primary care [13] also used small sample sizes. Nonetheless, we achieved significant differences according to our initial objectives. In such quasi-experimental studies, participant loss during recruitment and participation is common. A second limitation is the exclusion of chronic health problems as variables, except for excess weight and obesity, due to the intervention’s focus on health promotion and disease prevention among young adults. The study prioritised the key concepts of the middle-range theory of weight management [7], offering care plans for the identified nursing diagnoses of Readiness for Enhanced Nutrition (code 00163) and Readiness for Enhanced Exercise (code 00307). The third limitation is related to the significant change identified in the responses to the CADM by the IG. It could be considered whether these changes reflect better adherence to the diet or whether they are learned responses. This is a common limitation in any study that proposes to identify changes in habits through the completion of questionnaires. However, it is believed that these IG participants, who also improved their anthropometric parameters and physical activity, actually also increased their adherence to the Mediterranean diet, making it unlikely that they had memorized the answers three months after the initial assessment. The fourth limitation is the exclusively individual design of the intervention. Some patients expressed interest in participating in a group where they could have shared their achievements and challenges with their peers. Future replication of this intervention could include, in addition to larger samples, group activities to address this request and test its effectiveness.

## 5. Conclusions

Care4US+ has demonstrated to be an effective nurse-led intervention for weight management in individuals aged 18–45 years. It is based on information and communication technologies and its own theoretical framework, as well as a standardized system of nursing care languages for practice. An improvement in anthropometric measurements, adherence to a healthy diet and physical activity is observed.

## Figures and Tables

**Table 1 nutrients-16-02217-t001:** Description of sample characteristics by group (CG: n = 33, IG: n = 33).

Variables			Sample Characteristics	
			CG	IG
Sociodemographic	Sex % (n)	Women	66.7 (22)	72.7 (24)
		Men	33.3 (11)	27.3 (9)
	Age in years (mean, SD)		39.0 (5.6)	34.1 (8.8)
Anthropometric	Weight in kilograms (mean, SD)		89.1 (12.4)	91.1 (13.2)
	Abdominal circumference in centimetres (mean, SD)		103.9 (11.9)	104.8 (11.5)
	Body mass index (mean, SD)		31.4 (3.5)	32.4 (3.5)
Motivation	Baseline: 1 to 5, lowest to highest (mean, SD)		3.7 (0.5)	4.1 (0.7)
Dietary habits	CADM score (median, IQR)		7.0 (2.0)	7.0 (2.5)
Physical activity	Metabolic expenditure according to IPAQ (median, IQR)		1089.0 (3061.7)	975.0 (1993.1)

SD: standard deviation. CADM: Mediterranean Diet Adherence Questionnaire; IPAQ: International Physical Activity Questionnaire. IQR: inter-quartile range.

**Table 2 nutrients-16-02217-t002:** Analysis of differences in anthropometric characteristics, adherence to the Mediterranean diet, and physical activity level between the control group (CG: n = 33) and the intervention group (IG: n = 33) at baseline and final points.

Variables	Comparison of Results across Samples									
	CG					IG				
Anthropometric	Pre	95% CI	Post	95% CI	*p*-Value	Pre	95% CI	Post	95% CI	*p*-Value
Weight in kilograms: mean (SD)	89.1 (12.4)	84.8–93.4	90.2 (13.0)	85.7–94.7	0.056	91.1 (13.2)	86.4–95.7	88.8 (13.3)	84.2–93.4	0.003 **
Abdominal circumference in centimetres: mean (SD)	103.9 (11.9)	99.8–108.1	104.5 (13.4)	99.9–109.2	0.642	104.8 (11.5)	100.8–108.8	103.0 (11.9)	98.9–107.2	0.043 **
Body mass index: mean (SD)	31.4 (3.5)	30.2–32.6	31.8 (3.6)	30.6–33.1	0.046 **	32.4 (3.5)	31.2–33.7	31.6 (3.7)	30.4–32.9	0.005 **
	CG					IG				
	Pre		Post	Z	*p*-value	Pre		Post	Z	*p*-value
Dietary habits CADM score: median (IQR)	7.0 (2.0)		7.0 (2.0)	−1.693	0.090	7.0 (2.5)		9.0 (2.0)	−4.128	<0.001 **
Physical activity Metabolic expenditure according to IPAQ: median (IQR)	1089.0 (3061.7)		792.0 (2388.6)	−0.121	0.904	975.00 (1993.1)		2400.00 (5245.2)	−3.292	0.001 **

SD: standard deviation. CADM: Mediterranean Diet Adherence Questionnaire; IPAQ: International Physical Activity Questionnaire. IQR: inter-quartile range. 95% CI: 95% confidence intervals. Z: Wilcoxon Z-value. ** Statistically significant *p*-value.

**Table 3 nutrients-16-02217-t003:** Comparative analysis and significance of differences in percentage and number of correct responses for each item of the Mediterranean Diet Adherence Questionnaire (CADM).

CADM Items [Correct Response]	Comparison of Results across Samples					
	CG % (n)			IG % (n)		
	Pre	Post	*p*-Value	Pre	Post	*p*-Value
1. Do you use olive oil as your main cooking oil? [Yes]	84.8 (28)	87.9 (29)	1.000	78.8 (26)	93.9 (31)	0.125
2. How much olive oil do you consume in total per day? [Two or more tablespoons]	39.4 (13)	39.4 (13)	1.000	39.4 (13)	57.6 (19)	0.109
3. How many servings of vegetables do you eat per day? [Two or more per day]	30.3 (10)	36.4 (12)	0.727	33.3 (11)	54.5 (18)	0.039 **
4. How many pieces of fruit do you eat per day? [Three or more per day]	18.2 (6)	24.2 (8)	0.500	18.2 (6)	42.4 (14)	0.021 **
5. How many servings of red meat, hamburgers, sausages, or cold cuts do you eat per day? [Less than one per day]	81.8 (27)	81.8 (27)	1.000	66.7 (22)	90.9 (30)	0.021 **
6. How many servings of butter, margarine or cream do you consume per day? [Less than one per day]	93.9 (31)	93.9 (31)	1.000	93.9 (31)	100.0 (33)	0.500
7. How many carbonated and/or sugary drinks do you consume per day? [Less than one per day]	75.8 (25)	81.8 (27)	0.687	66.7 (22)	87.9 (29)	0.039 **
8. Do you drink wine? How many glasses of wine do you drink per week? [Three or more per week]	6.1 (2)	6.1 (2)	1.000	3.0 (1)	0.0 (0)	1.000
9. How many servings of pulses do you eat per week? [Three or more per week]	15.2 (5)	12.1 (4)	1.000	27.3 (9)	39.4 (13)	0.289
10. How many servings of fish or seafood do you eat per week? [Three or more per week]	12.1 (4)	21.2 (7)	0.375	9.1 (3)	24.2 (8)	0.063
11. How many times do you eat commercial baked goods such as biscuits, custards, cakes, or pastries per week? [Less than three per week]	51.5 (17)	54.5 (18)	1.000	60.6 (20)	72.7 (24)	0.344
12. How many times a week do you eat nuts? [Once or more per week]	48.5 (16)	54.5 (18)	0.625	63.6 (21)	78.8 (26)	0.180
13. Do you usually eat more chicken, turkey, or rabbit meat than beef, pork, hamburgers, or sausages? [Yes]	81.8 (27)	87.9 (29)	0.625	78.8 (26)	90.9 (30)	0.125
14. How many times a week do you eat cooked vegetables, pasta, rice, or other foodstuffs dressed with a tomato, garlic, onion, or leek sauce simmered with olive oil? [Twice or more per week]	27.3 (9)	36.4 (12)	0.250	42.4 (14)	48.5 (16)	0.687

** Statistically significant *p*-value.

**Table 4 nutrients-16-02217-t004:** Analysis of differences in anthropometric characteristics, adherence to the Mediterranean diet, and physical activity level between the control group (CG: n = 33) and the intervention group (IG: n = 33) by age groups at baseline and final points.

Variables	Comparison of Results across Samples									
	Age Range: 18–26 Years									
	CG (n = 2)					IG (n = 8)				
Anthropometric	Pre	95% CI	Post	95% CI	*p*-Value	Pre	95% CI	Post	95% CI	*p*-Value
Weight in kilograms: mean (SD)	96.4 (0.9)	95.2–97.6	97.0 (2.8)	93.0–101.0	0.742	95.6 (13.5)	86.1–105.1	93.4 (12.7)	84.4–102.4	0.132
Abdominal circumference in centimetres: mean (SD)	107.0 (12.7)	89.0–125.0	109.0 (15.6)	87.0–131.0	0.500	106.4 (13.4)	96.9–115.9	105.5 (10.6)	98.0–113.0	0.648
Body mass index: mean (SD)	35.2 (1.8)	32.6–36.5	35.5 (2.5)	31.9–39.1	0.729	32.9 (3.8)	30.2–35.6	32.1 (3.3)	29.7–34.5	0.128
	Pre		Post	Z	*p*-value	Pre		Post	Z	*p*-value
Dietary habits CADM score: Median (IQR)	7.0 (−)		7.0 (−)	0.000	1.000	5.5 (1.8)		8.0 (0.0)	−2.046	0.041 **
Physical activity Metabolic expenditure according to IPAQ: Median (IQR)	735.0 (−)		1455.0 (−)	−1.000	0.317	2005.2 (3366.0)		1906.5 (5663.1)	−1.782	0.075
	Age range: 27–35 years									
	CG (n = 9)					IG (n = 6)				
Anthropometric	Pre	95% CI	Post	95% CI	*p*-value	Pre	95% CI	Post	95% CI	*p*-value
Weight in kilograms: mean (SD)	84.2 (10.1)	77.5–90.9	86.6 (13.3)	77.7–95.5	0.251	87.0 (14.2)	75.4–98.6	82.1 (14.1)	70.6–93.6	0.016 **
Abdominal circumference in centimetres: mean (SD)	100.2 (7.1)	95.4–105.0	104.7 (15.1)	94.7–114.7	0.296	98.0 (11.4)	88.7–107.3	94.1 (13.9)	82.8–105.4	0.056
Body mass index: mean (SD)	30.1 (3.3)	27.9–32.3	30.9 (4.0)	28.3–33.5	0.239	31.1 (2.9)	28.8–33.4	29.4 (3.1)	26.9–31.9	0.014 **
	Pre		Post	Z	*p*-value	Pre		Post	Z	*p*-value
Dietary habits CADM score: median (IQR)	7.0 (1.5)		8.0 (1.5)	−1.000	0.317	7.0 (3.0)		9.5 (2.0)	−2.032	0.042 **
Physical activity Metabolic expenditure according to IPAQ: median (IQR)	1356.0 (3614.1)		594.0 (2077.5)	−0.943	0.345	1954.8 (6389.3)		4545.9 (9244.5)	−0.944	0.345
	Age range: 36–45 years									
	CG (n = 22)					IG (n = 19)				
Anthropometric	Pre	95% CI	Post	95% CI	*p*-value	Pre	95% CI	Post	95% CI	*p*-value
Weight in kilograms: mean (SD)	90.5 (13.3)	84.8–96.2	91.1 (13.3)	85.4–96.8	0.041 **	90.4 (13.1)	84.4–96.4	89.0 (13.1)	83.0–95.0	0.144
Abdominal circumference in centimetres: mean (SD)	105.2 (13.4)	99.5–110.9	104.1 (13.1)	99.0–109.7	0.201	106.3 (10.5)	101.5–111.1	104.8 (11.0)	99.8–109.8	0.210
Body mass index: mean (SD)	31.6 (3.4)	30.1–33.1	31.9 (3.5)	30.4–33.4	0.044 **	32.7 (3.6)	31.1–34.3	32.1 (3.8)	30.5–33.7	0.191
	Pre		Post	Z	*p*-value	Pre		Post	Z	*p*-value
Dietary habits CADM score: median (IQR)	7.0 (3.0)		7.0 (2.0)	−1.358	0.174	7.0 (2.0)		9.0 (2.0)	−3.071	0.002 **
Physical activity Metabolic expenditure according to IPAQ: median (IQR)	990.0 (2993.1)		954.6 (2441.4)	−0.549	0.583	792.0 (1739.4)		2395.2 (3889.2)	−2.792	0.005 **

SD: standard deviation. CADM: The Mediterranean Diet Adherence Questionnaire; IPAQ: The International Physical Activity Questionnaire. IQR: inter-quartile range. 95% CI: 95% confidence intervals. Z: Wilcoxon Z-value. ** Statistically significant *p*-value.

**Table 5 nutrients-16-02217-t005:** Frequency of responses regarding satisfaction perceived by the intervention group (IG).

Satisfaction-Related Questions	Response Rate % (n)					Median (IQR)
	Not at All Satisfied	Somewhat Satisfied	Moderately Satisfied	Very Satisfied	Fully Satisfied	
	1	2	3	4	5	
1. To what extent are you satisfied with the use and availability of the telehealth audiovisual resources employed when receiving the intervention?	3.0 (1)	18.2 (6)	24.2 (8)	30.3 (10)	24.2 (8)	4.0 (1.5)
2. How satisfied are you with the quality of the content of the intervention you received?	0 (0)	0 (0)	3.1 (1)	62.5 (20)	34.4 (11)	4.0 (1.0)
3. How satisfied are you overall with the educational intervention for the promotion of better dietary and physical activity habits?	0 (0)	0 (0)	3.1 (1)	59.4 (19)	37.5 (12)	4.0 (1.0)

IQR: inter-quartile range.

## Data Availability

The data presented in this study are available upon request from the corresponding author. The data are not publicly available due to privacy/ethical restrictions.

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
