# Peer review of "Effectiveness of a Nurse-Led Telehealth Intervention to Improve Adherence to Healthy Eating and Physical Activity Habits in Overweight or Obese Young Adults"

_nutrients, 2024, doi:10.3390/nu16142217_

Round 1

Reviewer 1 Report

Comments and Suggestions for Authors

Thank you for the opportunity review this quasi-experimental study with the general objective to improve the management of excess weight and obesity in a sample of adults visiting nurses’ offices in primary care facility following a telehealth-based educational intervention.

General – there’s a lot of acronyms that make it challenging to understand what the authors are referring to. Suggest adding a list of abbreviations/acronyms so reader isn’t searching through text to know what’s being discussed. Also suggest spelling out those that are regionally specific and used often in the text (i.e., telehealth instead of TH, and primary care facility instead of PCF).

The authors found significant changes in both physical measurements and behavior. However, it’s unclear if the changes in survey responses indicate behavior change or change in response. Please review tables where comparison groups are listed as CG and CI, and check articles from 2023.

Sampling MethodHow was the sample size was determined?

Materials and Methods –

·       How was this assessed? -- ‘To assess each potential participant’s readiness for change, they have to be in the preparation or action stage according to the Transtheoretical Model of Change by Prochaska and DiClemente, following initial assessment by the nurse’

·       Are these the questions listed in Table 5? Either put in the text or clarify where the reader can find the questions to which the author is referring. (Under -Construction and design of the data collection notebooks) ‘a final satisfaction assessment containing three questions related to the intervention, rated from 1 to 5, from lowest to highest satisfaction’

Results

·       The tables show ‘Sample Characteristics’ and ‘Comparison of results across samples’ for CG &CI, should it be CG and IG?

·       How do you know if the pre-post responses to the CADM represent a change in adherence vs. a change in response due to new knowledge?

References - A quick search using ‘telehealth intervention to promote weight loss and physical activity’ yielded references from 2023. Suggest authors do an updated literature search.

Author Response

We have attached, as a supplementary file, the language editing certificate.

Reviewer 2 Report

Comments and Suggestions for Authors

The manuscript is written well and demonstrates a good understanding of the subject matter. However, it is important to note that numerous articles have been published in this area, addressing various diseases. To enhance the quality and impact of the article, the authors should consider the following suggestions:
Inclusion and Exclusion Criteria: The manuscript lacks a detailed explanation of the inclusion and exclusion criteria for the study. This information is crucial as it defines the parameters for participant selection and ensures that the study is replicable. 
Sample Size: The current sample size appears to be quite small. A small sample size can limit the generalizability of the findings and may lead to biased or inconclusive results. It is essential to discuss the rationale behind the chosen sample size and consider conducting a power analysis to justify it. 
By addressing these points, the authors can improve the clarity, reliability, and overall quality of their manuscript.

Comments on the Quality of English Language

Minor English correction. 
